# Inter-Laboratory Comparison of Metabolite Measurements for Metabolomics Data Integration

**DOI:** 10.3390/metabo9110257

**Published:** 2019-10-31

**Authors:** Yoshihiro Izumi, Fumio Matsuda, Akiyoshi Hirayama, Kazutaka Ikeda, Yoshihiro Kita, Kanta Horie, Daisuke Saigusa, Kosuke Saito, Yuji Sawada, Hiroki Nakanishi, Nobuyuki Okahashi, Masatomo Takahashi, Motonao Nakao, Kosuke Hata, Yutaro Hoshi, Motohiko Morihara, Kazuhiro Tanabe, Takeshi Bamba, Yoshiya Oda

**Affiliations:** 1Division of Metabolomics, Medical Institute of Bioregulation, Kyushu University, 3-1-1 Maidashi, Higashi-ku, Fukuoka 812-8582, Japan; izumi@bioreg.kyushu-u.ac.jp (Y.I.); m-takahashi@bioreg.kyushu-u.ac.jp (M.T.); nakao@bioreg.kyushu-u.ac.jp (M.N.); k-hata@bioreg.kyushu-u.ac.jp (K.H.); 2Department of Bioinformatic Engineering, Graduate School of Information Science and Technology, Osaka University, 1-5 Yamadaoka, Suita, Osaka 565-0871, Japan; n-okahashi@ist.osaka-u.ac.jp; 3Institute for Advanced Biosciences, Keio University, 246-2 Mizukami, Kakuganji, Tsuruoka, Yamagata 997-0052, Japan; hirayama@ttck.keio.ac.jp; 4Laboratory for Metabolomics, RIKEN Center for Integrative Medical Sciences, 1-7-22 Suehiro-cho, Tsurumi-Ku, Yokohama, Kanagawa 230-0045, Japan; kazutaka.ikeda@riken.jp; 5Department of Lipidomics, Graduate School of Medicine, The University of Tokyo, 7-3-1 Hongo, Bunkyo-ku, Tokyo 113-0033, Japan; kita@m.u-tokyo.ac.jp (Y.K.); yoda@m.u-tokyo.ac.jp (Y.O.); 6Translational Science, Neurology Business Group, Eisai Co., Ltd., 5-1-3 Tokodai, Tsukuba, Ibaraki 300-2635, Japan; k2-horie@hhc.eisai.co.jp; 7Tohoku Medical Megabank Organization, Tohoku University, 2-1 Seiryo-machi, Aoba-ku, Sendai, Miyagi 980-8573, Japan; saigusa@m.tohoku.ac.jp; 8Division of Medical Safety Science, National Institute of Health Science, 3-25-26 Tonomachi, Kawasaki-ku, Kawasaki, Kanagawa 210-9501, Japan; saitok2@nihs.go.jp; 9RIKEN Center for Sustainable Resource Science, 1-7-22 Suehiro-cho, Tsurumi-ku, Yokohama, Kanagawa 230-0045, Japan; yuji.sawada@riken.jp; 10Research Center for Biosignal, Akita University, 1-1-1 Hondo, Akita-city, Akita 010-8543, Japan; hnakani@med.akita-u.ac.jp; 11Pharmacokinetic Research Laboratories, Ono Pharmaceutical Co., Ltd., 17-2 Wadai, Tsukuba, Ibaraki 300-4247, Japan; y.hoshi@ono.co.jp; 12Translational Research Laboratories, Ono Pharmaceutical Co., Ltd., 3-1-1 Sakurai Shimamoto-cho, Mishima-gun, Osaka 618-8585, Japan; morihara@ono.co.jp; 13Medical Solution Promotion Department, Medical Solution Segment, LSI Medience Corporation, 3-30-1, Shimura, Itabashi-ku, Tokyo 174-8555, Japan; tanabe.kazuhiro@mp.medience.co.jp

**Keywords:** metabolomics, relative quantification, method validation, inter-laboratory comparison, data integration, quality control sample

## Abstract

Background: One of the current problems in the field of metabolomics is the difficulty in integrating data collected using different equipment at different facilities, because many metabolomic methods have been developed independently and are unique to each laboratory. Methods: In this study, we examined whether different analytical methods among 12 different laboratories provided comparable relative quantification data for certain metabolites. Identical samples extracted from two cell lines (HT-29 and AsPc-1) were distributed to each facility, and hydrophilic and hydrophobic metabolite analyses were performed using the daily routine protocols of each laboratory. Results: The results indicate that there was no difference in the relative quantitative data (HT-29/AsPc-1) for about half of the measured metabolites among the laboratories and assay methods. Data review also revealed that errors in relative quantification were derived from issues such as erroneous peak identification, insufficient peak separation, a difference in detection sensitivity, derivatization reactions, and extraction solvent interference. Conclusion: The results indicated that relative quantification data obtained at different facilities and at different times would be integrated and compared by using a reference materials shared for data normalization.

## 1. Introduction

Metabolomics using mass spectrometry (MS) is a promising tool for the life science and biotechnology fields [1,2,3,4,5]. In recent years, relative quantification data produced mainly by the targeted metabolomics approach were integrated into large datasets for researches in medical biotechnology, precision medicine, cohort studies, epidemiological studies, and genome-wide association studies [6,7,8]. It is anticipated that a massive generation of reliable data by inter-laboratory collaborative research will promote data exchange and result in a larger accumulation of data over the next few decades. However, there are several problems regarding the quantification of metabolites that must be solved in order to share data obtained in different laboratories.

One of the biggest problems that must be addressed is that the targeted metabolome methods have not been well validated to guarantee the production of reliable and comparable data. For the case of drug metabolites analysis in biological samples, a draft guideline for method validation using blank matrix and standard materials has been published (International Council for Harmonization of Technical Requirements for Pharmaceuticals for Human Use) [9]. In targeted metabolomics, method validation using a similar approach seems to be quite a difficult task. This is because true values are unclear due to a lack of suitable blank matrix as well as many standard materials [10]. Therefore, reproducibility and reliability are always a matter of concern [11]. When peak separation is insufficient, the calculation of peak area or intensity becomes less accurate. Additionally, the certainty of identifying metabolites without standard substances is diminished [12].

Another problem is that the general gold standard approaches are lacking in metabolomics [13]. There are some activities regarding standardization of metabolome measurement; however, it is unlikely that all researchers conduct unified protocols, as multiple combinations of MS, including quadrupole mass spectrometry (QMS), time-of-flight mass spectrometry (TOFMS), quadrupole-time-of-flight mass spectrometry (QTOFS), triple quadrupole mass spectrometry (TQMS), quadrupole-Orbitrap mass spectrometry (Q Exactive), and quadrupole-Orbitrap-linear ion trap mass spectrometry (Orbitrap Fusion) can be used. Several separation techniques, such as gas chromatography (GC), liquid chromatography (LC), supercritical fluid chromatography (SFC), ion chromatography (IC), or capillary electrophoresis (CE), are also available for the investigation of metabolites in biological components.

Despite the diverse methods for relative quantitation, results that are essentially the same must be produced when identical samples are analyzed by different methods in different laboratories. When the results are similar, the relative quantitation results are probably close to the true value, and these methodologies seem to be reliable. Moreover, a global quality control or a reference material is available to normalize the datasets produced across various laboratories.

In the proteomic field, a multi-laboratory evaluation study at eleven sites around the world was conducted using an identical protocol for the sequential window acquisition of all theoretical mass spectra (SWATH-MS). They evaluated reproducibility for more than 4000 proteins in HEK293 cells and concluded that the quantitative characteristics were highly comparable across all sites [14]. In the metabolomics field, interlaboratory reproducibility for hydrophilic and hydrophobic metabolites (mainly amino acids and phospholipids) was tested through the analysis of human serum and plasma using Biocrates’ AbsoluteIDQ kit [15]. However, in the real world, different laboratories use different protocols on different devices. In this regard, community-based efforts reportedly improved the versatility of lipidomics. Consensus values of plasma lipids in the standard reference material (SRM) 1950‒metabolites in frozen human plasma were obtained through the LIPID MAPS consortium and the National Institute of Standards and Technologies (NIST) study group [16]. For instance, Bowden et al. conducted a thirty-one inter-laboratory study using different lipidomics workflows to SRM 1950 [17]. The results of absolute quantitation of plasma lipid concentration were different among these laboratories, suggesting that it is necessary to find the reasons behind these differences [18]. Issues that need to be addressed were found in the process of pre-analytics [19,20], analytics, and post-analytics [21,22]. Therefore, generally accepted guidelines for lipidomics across independent laboratories are required, and community-initiated approaches have been attempted for data harmonization among different instrumentation platforms [23].

In this study, we focused on the relative comparisons of 206 hydrophilic and 584 hydrophobic metabolite levels among samples and performed an inter-laboratory comparison study of metabolite measurements across 12 laboratories. One of the main purposes of this study was to verify the targeted metabolomics methods based on a majority rule. A comparison of relative quantitation data (HT-29/AsPc-1) showed that essentially identical results were obtained on more than half of the measured metabolites among laboratories and assay methods. Another purpose of the inter-laboratory study was to examine error shooting of metabolome analysis using a data review. The results confirmed that comparable data could be produced for more than half of the metabolites using different metabolome analysis methods, and several reasons leading to the incomparable results were successfully identified.

## 2. Results and Discussion

### 2.1. Experimental Design

In the present study, we performed relative quantitation of various metabolites in two identical samples using different targeted metabolomics methods (Figure 1). Two cell extract pools (samples) were prepared from HT-29 and AsPc-1 cell cultures. The two samples were then distributed to 12 participating laboratories throughout Japan. In each laboratory, the peak was identified for each metabolite and their relative signal intensity value (HT-29/AsPc-1) was measured according to their own routine analytical methods. We did not share any protocols or lists of target metabolites. Herein, an analytical method includes sample pretreatment, separation, detection, and data processing. A total of six analytical runs were performed for triplicate extracts of two cell lines in each method. The relative signal intensity data of metabolites were compiled to compare among laboratories. The integrated data were subsequently analyzed using statistical tests to evaluate whether similar results were produced by distinct analytical methods.

### 2.2. Data Acquisition and Integration

In this study, inconsistency in compound identifier (ID) usage became a problem during the data integrating task, because different compound names and IDs were used in different laboratories. Specifically, lipid structures have not been fully elucidated in many cases, and different names are given to MS peaks among laboratories due to the lack of standardized metabolite nomenclature [24]. Thus, we created a consensus list of compound IDs as well as a naming rule for lipidomics among participants in order to address this issue. For the hydrophobic metabolites (lipids) analysis, we targeted 21 lipid classes, including acylcarnitine (AC), cholesterol ester (ChE), ceramide (NS) (Cer(NS)), diacylglycerol (DG), free fatty acid (FA), hexose ceramide (NS) (HexCer(NS)), lysophosphatidic acid (LPA), lysophosphatidylcholine (LPC), lysophosphatidylethanolamine (LPE), lysophosphatidylglycerol (LPG), lysophosphatidylinositol (LPI), lysophosphatidylserine (LPS), monoacylglycerol (MG), phosphatidic acid (PA), phosphatidylcholine (PC), phosphatidylethanolamine (PE), phosphatidylglycerol (PG), phosphatidylinositol (PI), phosphatidylserine (PS), sphingomyelin (SM), and triacylglycerol (TG), with a variety of 20 FA side chains. Thus, the targeted lipid molecular species comprised 3390 compounds, including lysophospholipid positional isomers (e.g., LPA 16:0 (*sn*-1) and LPA 16:0 (*sn*-2)).

Identification of the hydrophilic metabolites, including amino acids, organic acids, sugar phosphates, nucleic acids, and coenzymes, was accomplished by each laboratory’s protocol such as by comparing the retention time (or migration time), MS and MS/MS spectra, and the multiple reaction monitoring (MRM) transition of the target metabolites in the samples with those of authentic standards analyzed under identical conditions. Identification of lipids was conducted based on the retention time, precursor ion, and the fragmentation patterns or specific MRM transitions of each molecule.

The metabolite extracts obtained from HT-29 and AsPc-1 cells with methanol were aliquoted and distributed among 12 laboratories in April 2018. These samples were to be used for metabolome analysis via routine measurements at participating laboratories. The list of measured metabolites and their relative amounts were returned by August 2018 and compiled into a dataset. The completed dataset consists of relative quantification results obtained using 15 methods for hydrophilic metabolites and nine methods for lipids (Table 1 and Table 2). It should be noted that ranges of measured metabolites were partly overlapped but different among laboratories.

By integrating the results from all analytical methods, a total of 206 hydrophilic metabolites and 584 hydrophobic metabolites were identified from the HT-29 and/or AsPc-1 cell extracts at least by one analytical method (Figure 2A and Table 3). Among these, a total of 148 hydrophilic metabolites and 285 hydrophobic metabolites in both HT-29 and AsPc-1 cell extracts were detected using two or more analytical methods (Figure 2B and Table 3). A total of 433 metabolites were identified by multiple methods, which corresponded to 54.8% of the total 790 metabolites. The number was much higher than that of previous report. For instance, it has been reported that 217 metabolites in total were identified by multiple methods [24]. These results demonstrate that the coverage of metabolomics could be greatly improved by sharing data among multiple laboratories.

### 2.3. Comparison of Analytical Methods for Relative Quantification

Relative ratios (HT-29/AsPc-1) of 206 hydrophilic metabolites obtained using 15 different methods and of 584 hydrophobic metabolites obtained using nine different methods are shown in Appendix A, respectively. Statistical significance of each metabolite between samples HT-29 and AsPc-1 was determined using a two-sided Student’s *t*-test (*p* < 0.05). In total, 104 hydrophilic metabolites and 199 hydrophobic metabolites were statistically significant between samples HT-29 and AsPc-1 among two or more methods (Table 3 and Appendix A). Among them, the HT-29/AsPc-1 values of five hydrophilic and seven hydrophobic metabolites were significantly changed to distinct directions among multiple methods (Table 3, Appendix A). For instance, the incorrect results were observed for acetylglycine (Acetyl-Gly), allantoin, dihydroxyacetone phosphate (DHAP), isocitric acid, (IsoCit), and lysine (Lys) as well as LPC 16:0 (*sn*-1), LPC 20:1 (*sn*-2), LPE 22:6 (*sn*-1), PE 16:0/18:0, PE 18:0/18:0, PG 16:0/18:1, and PI 18:0/20:2 (Appendix A). An obviously incorrect result was observed for only 4.0% (12/303) of metabolites in total, indicating a basic agreement of the relative amounts data among all analytical methods.

The one-way ANOVA using HT-29/AsPc-1 values (α = 0.05) shows that there is no difference between the 57 hydrophilic and 113 hydrophobic metabolites across the various methods (Table 3). The total of 170 metabolites corresponded to 39.2% for 433 metabolites identified from both samples by two or more methods. The results of the one-way ANOVA indicated that, when considering AsPc-1 cells as a global quality control sample in this study, the ratios of HT-29/AsPC-1 for 10 amino acids, including asparagine (Asn), aspartic acid (Asp), cysteine (Cys), glutamic acid (Glu), glycine (Gly), histidine (His), leucine (Leu), serine (Ser), threonine (Thr), and tyrosine (Tyr), were similar among multiple methods (Figure 3). However, the ratio of tryptophan (Trp) ranged from 0.93 to 1.11 in the six methods, but in one protocol was as high as 3.81 (Figure 3). Similar outliers were observed for 41 hydrophilic metabolites and 49 hydrophobic metabolites (Table 3 and Appendix A), likely due to errors in the analysis method as discussed below.

When ignoring one outlier in the data, the results improved such that HT-29/AsPc-1 levels were similar among 98 hydrophilic and 162 hydrophobic metabolites. The total 260 metabolites corresponded to 60.0% for 433 metabolites identified from both samples by two or more methods (Table 3). The values of these metabolites seem to be close to the true values because a similar result was obtained regardless of the difference in measurement method (Appendix A). Although the results cannot be compared directly, the analysis of plasma samples by the metabolomics platforms in Metabolon and Broad Institute demonstrated that 111 overlapped metabolites were correlated with each other (the median Spearman correlation was 0.79) [24]. Although the majority voting is not always correct and there are still many outliers, the results demonstrated that identical relative quantitation data was produced by the targeted metabolomics methods throughout various laboratories.

### 2.4. Community-Based Correction of Targeted Metabolome Analysis Methods

Data review of the inter-laboratory study also allowed for the troubleshooting of each metabolome analysis method. Since the dataset was large, we could review only a small part of this study. For instance, almost all the outlier values in the amino acid data including Gln, Phe, Pro, and Trp were produced by the methods using GC/MS (Figure 3). Among the GC/MS-based methods for hydrophilic metabolite analysis, including methods M, N, and O, the ratios of Gln and Pro determined by method N were slightly different from that of other methods. The result suggested that the derivatization reaction for Gln and Pro in method N might be insufficient. On the other hand, the relative quantification of Lys using method G (C18-LC/TQMS, MRM) was significantly different from the other seven methods (Figure 3). The possible reason was the co-elution with the isobaric ions because method G set similar MRM transitions for Lys (146.9 > 84.1) and Gln (147.0 > 84.1), and both compounds were not chromatographically separated.

Furthermore, several errors were derived from the insufficient separation of various structural isomers such as glucose and fructose. In this study, the relative quantification value of glucose was obtained from four laboratories (methods E, L, N, and O for hydrophilic metabolites). However, the value obtained from method L was an outlier (Figure 4A). The data review pointed out that the method L did not include fructose in the targeted metabolite list, suggesting that glucose and fructose were coeluted in the method L. A similar case was found in phospholipid analysis. For example, PI 38:2 contains PI 18:0/20:2 and PI 18:1/20:1. Both isomers were measured by methods C, E, and H for hydrophobic metabolites (Figure 4B). The relative values of PI 18:0/20:2 and PI 18:1/20:1 determined by methods C, E, and H were close to each other. On the other hand, methods A and D commonly employed the C18-LC separation and detected only PI 18:0/20:2. The relative values of PI 18:0/20:2 determined by methods A and D were significantly larger than that of other methods, suggesting that PI 18:0/20:2 and PI 18:1/20:1 was likely to be coeluted by methods A and D. Similarly, PE 34:2 primarily contains PE 16:1/18:1 and PE 16:0/18:2. Both isomers were measured by methods E and H that produced similar quantification values each other (Figure 4C). Another three facilities measured only PE 16:1/18:1 and produced incomparable values (Figure 4C). The data review found that different peaks having different retention times were identified in the method C. There results showed that these errors and misidentifications would not have been noticed unless it was checked using reference material. To minimize such error, strict thresholds for metabolic identification or peak integration, as well as standardization of data analysis, are desired. However, the work will require many compromises because one of the main objectives of metabolome analysis is to identify as many molecules as possible.

## 3. Materials and Methods

### 3.1. Cell Cultures

Human colon colorectal adenocarcinoma HT-29 cells and human pancreatic cancer AsPC-1 cells were obtained from the American Type Culture Collection (ATCC, Manassas, VA, USA). Both cells were cultured in RPMI 1640 medium (Nissui Pharmaceutical Co., Ltd., Tokyo, Japan) supplemented with 10% (*v*/*v*) fetal bovine serum, 100 unit mL^−1^ penicillin, 100 mg mL^−1^ streptomycin, and 0.25 mg mL^−1^ amphotericin B at 37 °C in a humidified atmosphere with 5% CO_2_.

### 3.2. Preparation of Cell Extract for Metabolite Quantification

Human HT-29 and AsPc-1 cells were seeded at a density of 4 × 10^6^ and 3 × 10^6^ cells/15 cm dish, respectively, and 31 dishes (one dish was used for cell count) were prepared for each sample. Each dish was cultured for 72 h. After the cultivation, the cells were washed with phosphate-buffered saline (PBS, Sigma-Aldrich, St Louis, MO, USA) and lysed in 5 mL (per dish) methanol. Thirty dishes were scraped and pooled for each sample. Finally, each sample cell number was adjusted to 4 × 10^6^ cells mL^−1^ by adding methanol. After mixing to obtain a homogenous solution, 350 µL of the sample were packaged at −80 °C and sent to individual laboratories.

### 3.3. Analytical Procedure

The metabolite extract was distributed among 12 laboratories and was analyzed using their own metabolome analysis methods, including 15 methods [25,26,27,28,29,30,31,32,33,34,35,36,37] for hydrophilic metabolites and nine methods [38,39,40,41,42,43,44] for hydrophobic metabolites. Here, an analysis method includes the pretreatment of samples (such as addition of internal standards, extraction, phase separation, clean up, and/or derivatization), chromatographic separation, MS detection, and data processing. From each six distributed samples (triplicate extracts of two cell lines), six pretreated samples were prepared. Metabolome data were obtained by six runs in total, from which signal intensity data were prepared. The details of each analytical method are described in the Appendix A.

### 3.4. Data Analysis

Data processing, including peak picking and metabolite identification, was conducted according to each laboratory’s specific method. The relative quantitative data (HT-29/AsPc-1) for each metabolite were obtained from the triplicate analyses. The details of data processing and normalization data for each analytical method are described in the Appendix A. The integrated data were subsequently analyzed based on a two-sided Student’s *t*-test or a one-way ANOVA to evaluate whether similar results were produced by multi-laboratory distinct analytical methods. Statistical significance of each metabolite between samples HT-29 and AsPc-1 was determined using a two-sided Student’s *t*-test (*p* < 0.05). The number of metabolites that showed HT-29/AsPc-1 levels in different directions among two or more methods was examined based on a two-sided Student’s *t*-test and a relative quantitative value of 1 (Appendix A). The number of metabolites with statistically similar HT-29/AsPc-1 levels was also examined among multiple methods using a one-way ANOVA analysis (*p* > 0.05). Subsequently, the number of metabolites that changed to a *p*-value of >0.05 by a one-way ANOVA when ignoring one outlier in the data showing a *p*-value of <0.05 by a one-way ANOVA was searched (Appendix A). All statistical analyses were performed using in-house scripts written in Python 3 using Numpy and Scipy modules.

## 4. Conclusions

In the present study, even if the results were obtained by different analytical methods, the results could be compared by relative quantification. This indicates that if global quality control samples or reference materials are prepared and shared for data normalization, data obtained at different facilities and at different times may be integrated and compared. On the contrary, each analytical method that gave similar results for a certain substance seemed to be accurate. However, it is difficult to generalize this, and it seems that it can be applied only to specific metabolites. Of course, if the matrix (i.e., serum, plasma, urine, etc.) is different, the assay must be revalidated for each metabolite. The reason why the results differed among analyses was not only that other peaks (i.e., isobaric or isomeric ions) were overlapping, but also because the peaks were identified incorrectly, or the sensitivity was poor and the noise was significantly impacted. For the former, it could be improved by using a standard material and for the latter by specifying a lower limit of quantification. In this study, relative values were often very different, but if these variations could be corrected, relative quantitative data obtained from different laboratories could be integrated for new biological findings by a large dataset that cannot be achieved by one facility. It can also be a means to verify the certainty of new analytical methods.

## Figures and Tables

**Figure 1 metabolites-09-00257-f001:**
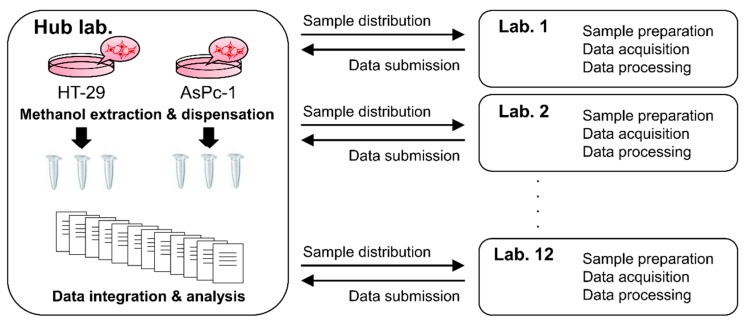
Experimental design in this study. Two crude extract pools were prepared from HT-29 and AsPc-1 cell cultures. The two samples were dispensed and distributed to 12 participating laboratories. In each laboratory, the peak was identified for each metabolite and their relative signal intensity value (HT-29/AsPc-1) was measured according to their sample pretreatment, separation, detection, and data processing methods. Data were compiled and analyzed using statistical tests.

**Figure 2 metabolites-09-00257-f002:**
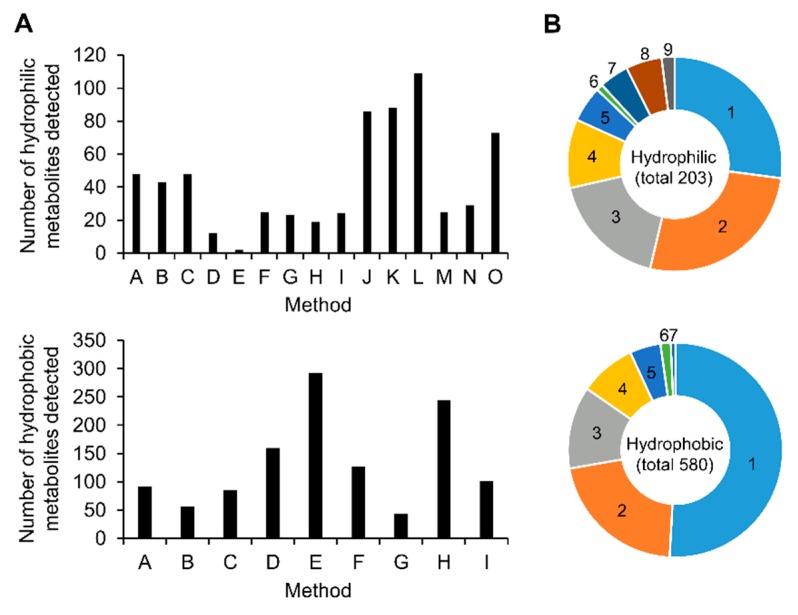
Number of metabolites detected by each analytical method (**A**) and percentage of metabolites commonly measured by the multiple methods (**B**). In total 203 hydrophilic metabolites and 580 hydrophobic metabolites were identified in both HT-29 and AsPc-1 cell extracts. Among them, 148 hydrophilic metabolites and 285 hydrophobic metabolites were measured across multiple methods.

**Figure 3 metabolites-09-00257-f003:**
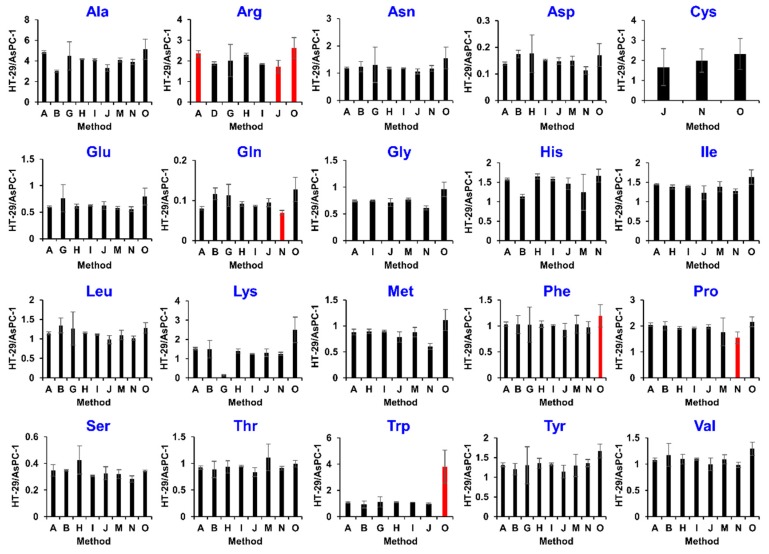
Inter-laboratory comparison of relative quantification for amino acids. Red bars indicate outliers based on a one-way ANOVA analysis (α = 0.05). Values are presented as the mean ± SD obtained from triplicate experiments.

**Figure 4 metabolites-09-00257-f004:**
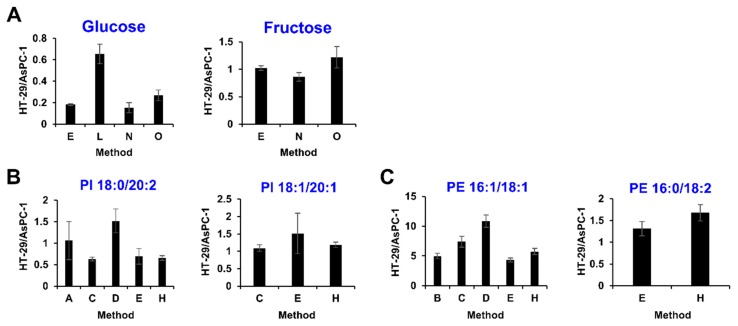
The effect of isomer discrimination on comparison of relative determinations for hexoses (**A**), PI 38:2 (**B**), and PE 34:2 (**C**). Values are presented as the mean ± SD obtained from triplicate experiments.

**Table 1 metabolites-09-00257-t001:** Information on analytical methods used for hydrophilic metabolites.

Method ID	Lab ID	Analytical Method	Analytical Mode	Targeted Hydrophilic Metabolites	Ref.
A	1	CE-TOFMS (cation mode)	Scan (positive)	Amino acids, Bases, Nucleosides, Amines, etc.	[25]
B	2	CE-TOFMS (cation mode)	Scan (positive)	Amino acids, Bases, Nucleosides, Amines, etc.	[25]
C	1	CE-TOFMS (anion mode)	Scan (negative)	Organic acids, Sugar phosphates, Nucleotides, etc.	[26]
D	2	CE-TOFMS (anion mode)	Scan (negative)	Organic acids, Sugar phosphates, Nucleotides, etc.	[27]
E	1	CE-TQMS (anion mode)	MRM (negative)	Monosaccharides	[28]
F	2	C18-LC/QTOFMS	Scan (positive/negative)	Amino acids, Organic acids, Nucleotides, etc.	[29]
G	3	C18-LC/TQMS	MRM (positive/negative)	Amino acids, Organic acids, Nucleotides, etc.	[30]
H	4	C18-LC/TQMS	MRM (positive)	Amino acids	[31]
I	5	Derivatization and C18-LC/TQMS	MRM (positive)	Amino acids, etc.	[32]
J	6	PFPP-LC/Q Exactive	Scan (positive/negative)	Amino acids, Bases, Nucleosides, Amines, Organic acids, etc.	[33]
K	1	Capillary-IC/Q Exactive	Scan (negative)	Organic acids, Sugar phosphates, Nucleotides, etc.	[34]
L	6	IC/Q Exactive	Scan (positive/negative)	Organic acids, Sugar phosphates, Nucleotides, etc.	[34]
M	7	Derivatization and GC/QMS	SIM	Amino acids, Organic acid, etc.	[35]
N	6	Derivatization and GC/QMS	Scan	Amino acids, Organic acid, etc.	[36]
O	8	Derivatization and GC/TQMS	MRM	Amino acids, Organic acid, Base, Nucleosides, etc.	[37]

PFPP, pentafluorophenylpropyl; and SIM, selected ion monitoring.

**Table 2 metabolites-09-00257-t002:** Information on analytical methods used to analyze hydrophobic metabolites.

Method ID	Lab ID	Analytical Method	Analytical Mode	Targeted Lipids	Ref.
A	9	C18-LC/QTOFMS	Scan (positive/negative)	ChE, Cer(NS), DG, FA, HexCer(NS), LPC, LPE, LPI, LPS, PC, PE, PG, PI, PS, SM	[38]
B	5	C8-LC/Q Exactive	Scan (positive)	AC, HexCer(NS), LPC, LPE, LPS, PC, PE, SM	[39]
C	5	C8-LC/Q Exactive	Scan (negative)	LPE, LPG, LPI, PC, PE, PG, PI, PS	[39]
D	10	C18-LC/Orbitrap Fusion	Scan (positive/negative)	AC, ChE, Cer(NS), DG, HexCer(NS), LPC, LPE, LPI, LPS, PC, PE, PG, PI, PS, SM, TG	[40]
E	11	C8-LC/TQMS	MRM (positive/negative)	LPC, LPE, LPI, LPS, PC, PE, PI, PS, SM	‒
F	12	C18-LC/TQMS	MRM (positive)	ChE, Cer(NS), LPA, LPC, LPE, LPG, LPI, LPS, MG	[41]
G	8	C18-LC/TQMS	MRM (positive/negative)	LPA, LPC, LPE, LPG, LPI, LPS	[42]
H	6	Diethylamine-SFC/TQMS	MRM (positive/negative)	Cer(NS), DG, HexCer(NS), LPC, LPE, MG, PA, PC, PE, PG, PI, PS, SM	[43]
I	6	C18-SFC/TQMS	MRM (positive/negative)	ChE, FA, TG	[44]

**Table 3 metabolites-09-00257-t003:** Summary of the dataset.

	Hydrophilic Metabolite	Hydrophobic Metabolite	Total
1. Number of identified metabolites from HT-29 and/or AsPc-1 samples by at least one analytical method	206	584	790
2. Number of identified metabolites from both samples by two or more methods	148	285	433
3. Number of metabolites that were statistically significant between samples HT-29 and AsPc-1 among multiple methods based on a two-sided Student’s *t*-test (α = 0.05)	104	199	303
4. Number of metabolites that showed HT-29/AsPc-1 levels in different directions among methods based on a two-sided Student’s *t*-test (α = 0.05) and a relative quantitative value of 1	5	7	12
5. Number of metabolites that were statistically similar HT-29/AsPc-1 levels among multiple methods using a one-way ANOVA (α = 0.05)	57	113	170
6. Number of metabolites that were statistically similar HT-29/AsPc-1 levels among multiple methods ignoring one outlier method using a one-way ANOVA (α = 0.05)	98	162	260

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
