# Peer review of "Inter-Laboratory Comparison of Metabolite Measurements for Metabolomics Data Integration"

_metabolites, 2019, doi:10.3390/metabo9110257_

Round 1
Reviewer 1 Report
These types of studies are always very important. Being able to investigate reproducibility of results across multiple labs and multiple platforms generates a powerful dataset for future analysis and allows us to understand the limitations of data integration between similar datasets which were separately obtained.
I particularly liked the depth of the investigation into the differences seen between the anlaytical techniques.
However the presentation of the results around Table 3 should be improved. I was very confused by the categories in table 3. More explanation of these should be included in the methods. I was especially confused as to why the number of metabolites which were deemed similar between methods dropped when excluding one outlying method (comparing the numbers in row 5 and row 6). I would also like further clarification of the phrase "significantly changed to distinct directions" which is used in the main text near to where Table 3 is discussed.
Minor points:
Line 261 - is desired, should be 'are desired' as we are discussing multiple desriable things.
Line 277 - 'Both' should be deleted.
Author Response
Response to the comments from Reviewer 1
[COMMENTS FOR THE AUTHOR]
These types of studies are always very important. Being able to investigate reproducibility of results across multiple labs and multiple platforms generates a powerful dataset for future analysis and allows us to understand the limitations of data integration between similar datasets which were separately obtained.
[Response]
Authors appreciate the reviewer’s useful comments to our manuscript. According to your comments and suggestions, the revised points were described below. The revision parts have been indicated with red-colored letters in our revised manuscript.
[Comment 1]
I particularly liked the depth of the investigation into the differences seen between the anlaytical techniques. However the presentation of the results around Table 3 should be improved. I was very confused by the categories in table 3. More explanation of these should be included in the methods. I was especially confused as to why the number of metabolites which were deemed similar between methods dropped when excluding one outlying method (comparing the numbers in row 5 and row 6). I would also like further clarification of the phrase "significantly changed to distinct directions" which is used in the main text near to where Table 3 is discussed.
[Response]
According to your kind comments, we revised the main text (2.3. Comparison of Analytical Methods for Relative Quantification and 3.4. Data Analysis) and Table 3.
Please see the revised manuscript at pages 6-7, lines 190-220 and page 10, lines 304-314.
Minor points:
[Comment 2]
Line 261 - is desired, should be 'are desired' as we are discussing multiple desriable things.
[Response]
According to your kind suggestion, we revised the word ‘is desired’ to ‘are desired’..
Please see the revised manuscript at page 8, line 264.
[Comment 3]
Line 277 - 'Both' should be deleted.
[Response]
According to your kind suggestion, we deleted the word ‘Both’.
Please see the revised manuscript at page 9, line 282.
Reviewer 2 Report
The main question addressed by the article is to make targeted metabolomics data (using mass spectrometry) universally comparable removing the differential factors such as method, analysis techniques, matrix effect etc.
The topic is a consensus need in the metabolomics mass spectrometry community. Therefore, it is not novel in itself but worthy of investigating.
This is systematic and through test of protocol to standardize targeted metabolomics data across labs and different protocols to help strengthen the quantitation of metabolomics data. Although this does not give a concrete answer about what specific standards to be used, it does sets the foundation of an approach that can help, with further experimental refinements, quantitation values of metabolites universal across different mass spectrometry platforms, protocols, sample preparation steps etc.
The paper is written well. It is easy to read and understand with proper level of attention given to explain data using graphs and details about methods used (provided in supplementary materials)
The conclusion is consistent with the arguments presented in the discussion section. However, it could have been proved experimentally by following up with another set of experiments in line with the suggestion of the conclusion section.
Yes, the article addresses the question of making the data universal cross different lab and protocol setups by using universal standards and defining lower limit of detection.
L303-305: This could easily be put to practice for validity check by providing the same labs with another batch of samples and an universal standard. Any reasons why this was not done?
Author Response
Response to the comments from Reviewer 2
[COMMENTS FOR THE AUTHOR]
The main question addressed by the article is to make targeted metabolomics data (using mass spectrometry) universally comparable removing the differential factors such as method, analysis techniques, matrix effect etc.
The topic is a consensus need in the metabolomics mass spectrometry community. Therefore, it is not novel in itself but worthy of investigating.
This is systematic and through test of protocol to standardize targeted metabolomics data across labs and different protocols to help strengthen the quantitation of metabolomics data. Although this does not give a concrete answer about what specific standards to be used, it does sets the foundation of an approach that can help, with further experimental refinements, quantitation values of metabolites universal across different mass spectrometry platforms, protocols, sample preparation steps etc.
The paper is written well. It is easy to read and understand with proper level of attention given to explain data using graphs and details about methods used (provided in supplementary materials)
[Response]
Authors appreciate the reviewer’s kind and useful comments to our manuscript.
[Comment 1]
The conclusion is consistent with the arguments presented in the discussion section. However, it could have been proved experimentally by following up with another set of experiments in line with the suggestion of the conclusion section.
Yes, the article addresses the question of making the data universal cross different lab and protocol setups by using universal standards and defining lower limit of detection.
L303-305: This could easily be put to practice for validity check by providing the same labs with another batch of samples and an universal standard. Any reasons why this was not done?
[Response]
The purpose of this study was not to compare differences in sensitivity among different laboratories and assay methods, but to compare the accuracy of relative quantitative values for metabolites using the daily routine protocols of each laboratory.
Since the metabolites detected in common in each method were targeted for evaluation, the importance of measuring standards was weak. As you have pointed out, it is important to perform sensitivity verifications with standards and spike-recovery-tests to investigate and verify the causes of metabolites that vary in relative quantitative values among different facilities and methods.
In the next research, we would like to make a sufficient research plan based on this reflection and proceed with research toward the integration of metabolomics data.
Thank you again for your important comments.
Reviewer 3 Report
The authors of this paper present an investigation into how targeted metabolomics can be compared across different laboratories which is an important question if the field of metabolomics is supposed to advance to the level of other omics. While the paper overall is well introduced and addresses this important question, I feel that the authors could have done more to present and interpret the results. It lacks in particular when it comes to comparing classes of metabolites and the methods used to analyze them that not only showed the best agreement when compared across different laboratories but also within, i.e. a “within-lab variation” when several methods were used and a “between-lab comparison” for the same class of metabolites.
Major Comments
Introduction
It is not clear from the introduction which metabolites are measured in the targeted analyses. The authors spend a great amount of text and the readers time on highlighting the efforts that have gone into lipidomics, but do not spend any time on e.g. amino acids or other targeted analyses. The reader has to assume that in the end it will become clear which metabolites in particular are going to be compared across laboratories. It is not mentioned anywhere in the introduction how many metabolites were analysed in total.Results
Line 220: The authors state that “the average was 0.79”. I would like to know what this average refers to. Is it the average correlation coefficient? The average relative ratio? One wonders. It is completely unclear which laboratories used the same or differing methods. A summary table detailing how many labs use the same methods method would greatly help comparability. Further, a comparison across laboratories using the same methods for the measurements would add to the text and the paper. It is not clear in the results section how the outlier was determined to be an outlier. A sentence detailing this would aid the reader. While the authors do make a distinction between hydrophobic and hydrophilic metabolites, I would like to see an additional split into the different classes of metabolites and how these compare across the different laboratories. A suggested split could be according to the different classes separated out in Table 1 and 2, e.g., “Amino acids and their related metabolites”, “Comprehensive anionic polar metabolites”, etc.Minor comments
Results
The number of brackets in some of the results paragraphs is way too high and hinders readability. Please remove at least 50% of the brackets to improve the flow of the text. The English used in Section 2.4 is below the high level of English set as standard in the rest of the paper.Author Response
Response to the comments from Reviewer 3
[COMMENTS FOR THE AUTHOR]
The authors of this paper present an investigation into how targeted metabolomics can be compared across different laboratories which is an important question if the field of metabolomics is supposed to advance to the level of other omics. While the paper overall is well introduced and addresses this important question, I feel that the authors could have done more to present and interpret the results. It lacks in particular when it comes to comparing classes of metabolites and the methods used to analyze them that not only showed the best agreement when compared across different laboratories but also within, i.e. a “within-lab variation” when several methods were used and a “between-lab comparison” for the same class of metabolites.
[Response]
Authors appreciate the reviewer’s useful comments to our manuscript. According to your comments and suggestions, the revised points were described below. The revision parts have been indicated with red-colored letters in our revised manuscript.
Major Comments
[Comment 1]
Introduction
It is not clear from the introduction which metabolites are measured in the targeted analyses. The authors spend a great amount of text and the readers time on highlighting the efforts that have gone into lipidomics, but do not spend any time on e.g. amino acids or other targeted analyses. The reader has to assume that in the end it will become clear which metabolites in particular are going to be compared across laboratories. It is not mentioned anywhere in the introduction how many metabolites were analysed in total.
[Response]
According to your suggestions, we revised the sentences in introduction at page 2, lines 92-93 and page 3, line 106.
[Comment 2]
Results
Line 220: The authors state that “the average was 0.79”. I would like to know what this average refers to. Is it the average correlation coefficient? The average relative ratio? One wonders. It is completely unclear which laboratories used the same or differing methods.
[Response]
According to your suggestion, we revised the sentences at page 7, line 224.
[Comment 3]
A summary table detailing how many labs use the same methods would greatly help comparability. Further, a comparison across laboratories using the same methods for the measurements would add to the text and the paper.
[Response]
Unfortunately, none of the analytical methods (pretreatment protocols and instrumental analysis) used in this study were identical. According to your kind comments, we added lab ID and analytical mode information to Tables 1 and 2.
Please see the revised manuscript at pages 4-5.
[Comment 4]
It is not clear in the results section how the outlier was determined to be an outlier. A sentence detailing this would aid the reader.
[Response]
According to your comment, we revised the main text (2.3. Comparison of Analytical Methods for Relative Quantification and 3.4. Data Analysis) and Table 3.
Please see the revised manuscript at pages 6-7, lines 190-220 and page 10, lines 304-314.
[Comment 5]
While the authors do make a distinction between hydrophobic and hydrophilic metabolites, I would like to see an additional split into the different classes of metabolites and how these compare across the different laboratories. A suggested split could be according to the different classes separated out in Table 1 and 2, e.g., “Amino acids and their related metabolites”, “Comprehensive anionic polar metabolites”, etc.
[Response]
According to your kind comments, we revised the ‘Targeted hydrophilic metabolites information’ in Table 1. Please see the revised manuscript at pages 4-5.
Minor comments
[Comment 6]
Results
The number of brackets in some of the results paragraphs is way too high and hinders readability. Please remove at least 50% of the brackets to improve the flow of the text.
[Response]
According to your kind comments, we revised the manuscript.
Please see the revised manuscript at pages 6-9, lines 172-266.
[Comment 7]
The English used in Section 2.4 is below the high level of English set as standard in the rest of the paper.
[Response]
According to your kind comments, we revised the manuscript in Section 2.4.
Please see the revised manuscript at pages 7-9, lines 230-266.
Round 2
Reviewer 3 Report
The authors have responded to all my comments well.